# Haloperidol Instigates Endometrial Carcinogenesis and Cancer Progression by the NF-κB/CSF-1 Signaling Cascade

**DOI:** 10.3390/cancers14133089

**Published:** 2022-06-23

**Authors:** Jung-Ying Chiang, Fu-Ju Lei, Huan-Jui Chang, Sung-Tai Wei, Chi-Chung Wang, Yen-Chih Huang, Hwai-Lee Wang, Chi-Fen Chuang, Shu-Yu Hu, Chia-Hung Hsieh

**Affiliations:** 1Graduate Institute of Biomedical Sciences, China Medical University, Taichung 404328, Taiwan; dragonchris@gmail.com (J.-Y.C.); zaireeriaz@gmail.com (S.-T.W.); junnasuke001@gmail.com (H.-L.W.); jifenchuang@gmail.com (C.-F.C.); estherhu9205@gmail.com (S.-Y.H.); 2Department of Neurosurgery, China Medical University Hsinchu Hospital, Hsinchu 302056, Taiwan; 3Graduate Institute of Clinical Medical Sciences, China Medical University, Taichung 404328, Taiwan; bettylei.fj@gmail.com; 4School of Medicine, Chung Shan Medical University, Taichung 402306, Taiwan; s0801093@gm.csmu.edu.tw; 5Department of Neurosurgery, China Medical University and Hospital, Taichung 404327, Taiwan; 6Graduate Institute of Biomedical and Pharmaceutical Science, Fu Jen Catholic University, New Taipei 242062, Taiwan; 075006@mail.fju.edu.tw; 7Department of Medical Imaging, China Medical University Hospital, Taichung 404327, Taiwan; arvin32.huang@gmail.com; 8Department of Medical Research, China Medical University Hospital, Taichung 404327, Taiwan; 9Department of Biomedical Informatics, Asia University, Taichung 413305, Taiwan

**Keywords:** haloperidol, endometrial cancer, endometrial hyperplasia, nuclear factor kappa B, colony-stimulating factor 1, carcinogenesis

## Abstract

**Simple Summary:**

Haloperidol, a typical antipsychotic, is widely used in schizophrenia and palliative care of cancer; however, the role and impact of chronic haloperidol treatment in endometrial cancer (EC) development are unclear. Here, we showed that haloperidol is a carcinogenic compound capable of inducing endometrial hyperplasia and promoting EC progression in rodents. Mechanistically, haloperidol stimulates the production of colony-stimulating factor 1 (CSF-1) on tumor cells by activating nuclear factor kappa B (NF-κB), and its downstream autocrine oncogenic CSF-1 receptor signaling contributes to this carcinogenesis. Furthermore, we demonstrated that the use of haloperidol is associated with increased EC-specific mortality in EC patients. Overall, these findings highlighted that physicians should be cautious about the use of haloperidol in female patients.

**Abstract:**

Haloperidol is a routine drug for schizophrenia and palliative care of cancer; it also has antitumor effects in several types of cancer. However, the role of haloperidol in endometrial cancer (EC) development is still unclear. Here, we show that chronic haloperidol treatment in clinically relevant doses induced endometrial hyperplasia in normal mice and promoted tumor growth and malignancy in mice with orthotopic EC. The pharmacokinetic study indicated that haloperidol highly accumulated in the uterus of mice. In vitro studies revealed that haloperidol stimulated the cellular transformation of human endometrial epithelial cells (HECCs) and promoted the proliferation, migration, and invasion of human endometrial carcinoma cells (HECCs) by activating nuclear factor kappa B (NF-κB) and its downstream signaling target, colony-stimulating factor 1 (CSF-1). Gain of function of CSF-1 promotes the cellular transformation of HEECs and the malignant progression of HECCs. Moreover, blockade of CSF-1 inhibited haloperidol-promoted EC progression in vitro and in vivo. A population-based cohort study of EC patients further demonstrated that the use of haloperidol was associated with increased EC-specific mortality. Collectively, these findings indicate that clinical use of haloperidol could potentially be harmful to female patients with EC.

## 1. Introduction

Haloperidol is a butyrophenone-derivative antipsychotic that enacts its potency by antagonizing dopamine receptor D2 (DRD2) in the limbic system [1]; it also has other blocking activities for noradrenergic, cholinergic, and histaminergic receptors. Based on the complexity of its non-selective binding ability, various adverse effects of haloperidol have been reported [2]. Nevertheless, it is still commonly used in schizophrenia to mitigate hallucinations and delusions. Furthermore, it has numerous FDA-approved or off-label clinical applications such as Tourette syndrome, hyperactivity, acute mania, and intractable hiccups. Moreover, it is widely used in palliative care of cancer patients due to its remedies for agitated delirium, nausea, and vomiting [3]. 

Emerging evidence shows that haloperidol has antitumor effects in several cancer types. Haloperidol restrains cell proliferation and initiates apoptosis in human melanoma, colon cancer, and breast cancer cells by binding to sigma receptors [4,5,6], which regulate tumor growth, cell proliferation, and tumor aggressiveness [7]. Moreover, haloperidol derepresses dual-specificity phosphatase 6 (DUSP6) by intronic demethylation and inhibits the proliferation of human pancreatic cancer cells [8]. In glioblastoma, haloperidol inhibits the formation of spheroids in vitro and tumor growth in vivo by targeting the DRD2 [9,10]. Similar effects and mechanisms are also observed in the orthotopic xenograft mice models of the pancreatic adenocarcinoma [11]. These findings suggest a potential role for haloperidol in anti-cancer therapy.

In contrast to its antitumor properties, haloperidol could be an insidious agent for initiating and promoting cancer via elevating serum prolactin, a hormone/cytokine. Haloperidol and other antipsychotics increase circulating prolactin levels by antagonizing DRD2 in the pituitary gland [12,13]. Furthermore, emerging evidence shows that prolactin has protumor properties in breast, prostate, colorectal, liver, ovarian, and endometrial cancers [14]. Recently, several epidemiological studies showed that prolactin-elevating antipsychotics increase the risk of breast cancer and breast cancer-specific mortality [15,16,17]. Therefore, more attention is needed to dissect the role of antipsychotics in cancer progression.

Endometrial cancer (EC) is the most prevalent gynecological malignancy worldwide [18,19]. A case-control study reported that the use of antipsychotics, which include sulpiride, levomepromazine, chlorpromazine, haloperidol, risperidone, pimozide, and clocapramine, is a risk factor for EC in the premenopausal women [20]. This association is linked to elevated prolactin in antipsychotic users. However, the sample size in this study is small, and the risk factor for individual antipsychotics has not been analyzed. Additionally, the role and mechanism of haloperidol in EC are still unclear. In the present study, we provide compelling evidence that haloperidol gives rise to endometrial carcinogenesis and promotes endometrial progression directly through the induction of nuclear factor kappa B (NF-κB) and its subsequent signaling target, colony-stimulating factor 1 (CSF-1).

## 2. Materials and Methods

### 2.1. Ethics Statement

We performed all animal studies after obtaining permission from the Institutional Animal Care and Use Committee (IUCAC) at China Medical University, Taiwan (CMUIACUC-2016-409). The IUCAC evaluates the proper conduct of animal studies following the Animal Protection Act of Taiwan, which complies with the principle of 3Rs: replacement, reduction, and refinement. The Ethical Review Board of China Medical University (Taiwan) has approved the clinical studies (CMUH105-REC3-083) in accordance with the Declaration of Helsinki, and the subjects have provided their informed consent.

### 2.2. Compounds and Antibodies

Haloperidol (Sigma-Aldrich, Burlington, MA, USA) was dissolved in normal saline; it was dosed in mice at 2 mg/kg or 1 mg/kg for in vivo studies and was mixed in the medium at different concentrations (0–100 μM) for in vitro studies. BLZ945 (Chemshuttle, Burlingame, CA, USA) was dissolved in 20% of Captisol^®^; it was dosed in mice at 200 mg/kg for in vivo studies and mixed in the medium to a final concentration of 500 nM for in vitro studies. SN50 (Sigma-Aldrich) and BAY 11-7082 (Sigma-Aldrich) were dissolved in double-distilled water and DMSO, respectively. The final concentrations are 50 μg/mL for SN50 and 2.5 μM for BAY 11-7082. Human CSF-1 recombinant protein (rCSF-1; R&D Systems, Minneapolis, MN, USA) was dissolved in double-distilled water to reach a 10 ng/mL final concentration for in vitro studies. The following primary antibodies were employed for Western blotting: human anti-CSF-1 antibody (diluted 1:550; cod. CF806567; Thermo Fisher Scientific, Waltham, MA, USA), human anti-phospho-Ser536-p65 (diluted 1:500; cod. MA5-15160; Thermo Fisher Scientific), and anti-β-actin antibody (diluted 1:10,000; cod. A2228; Sigma-Aldrich). CSF-1 neutralizing antibody (100 ng/mL; cod. AB-216-NA; R&D Systems) was used for antagonizing CSF-1.

### 2.3. Cell Culture

The human endometrial carcinoma cells (HECCs), AN3CA, HEC1A, and KLE, were generously supplied by Professor Ming-Ching Kao (China Medical University, Taichung, Taiwan). All HECCs were acquired from the American Type Culture Collection (ATCC) and maintained in the medium specified by the organization’s guidelines (DMEM (Life Technologies, Carlsbad, CA, USA) with 10% fetal bovine serum (FBS), 10 mM HEPES, and 1% penicillin-streptomycin). Primary human endometrial epithelial cells (HEECs) were isolated from endometrial biopsy samples provided by two healthy donors of reproductive age in the luteal phase according to the published protocol [21]. HEECs were cultured in an estrogen-free medium (75% DMEM without phenol red; 25% MCDB-105) with 10% human albumin, 5 mg/mL insulin, and antibiotics. All cells were cultivated in a CO_2_ incubator at 37 °C with humidified air (5% CO_2_ plus 20% O_2_ with N_2_ as the balance).

### 2.4. Establishment of Tumor Reporter Cells

pGreenFire1-SFFV [22], a pGreenFire1-mCMV vector backbone (System Biosciences, Palo Alto, USA) subcloned with the spleen focus-forming virus (SFFV) promoter, was utilized as the lentiviral envelope vector for constructing HEC1A that bear dual optical reporter genes, green fluorescence protein (GFP) and luciferase (Luc), driven by the SFFV promoter. The 293T cells were co-transfected with the pGreenFire1-SFFV, the packaging plasmid pCMVΔR8.91, and the envelope plasmid pMD.G by jetPRIME^®^ (Polyplus-transfection^®^ SA, Illkirch, France) transfection reagent. At 48 h after transfection, the viral particles in the culture medium were harvested and stored at −80 °C until use. HEC1A cells were infected with lentivirus in the presence of Polybrene (Sigma-Aldrich) at a final concentration of 8 μg/mL for 24 h and subject to ampicillin selection. For stable expression of dual optical reporter genes, infected HEC1A underwent one round of fluorescence-activated cell sorting (BD FACSAria^TM^ III; BD Biosciences, Franklin Lakes, NJ, USA). HEC1A bearing pGreenFire1-SFFV was therefore termed HEC1A-lucGFP.

### 2.5. Western Blot Assay (WB)

Proteins in HECCs were extracted using RIPA buffer, denatured in RIPA buffer and Laemmli buffer under 100 °C for 5 min, loaded (30 μg per well) into 5% stacking gel, and ran at 50 V in the stacking gel, whereas 100 V in the running gel (12% for CSF-1; 8% for phospho-Ser536-p65), and transferred onto the polyvinylidene difluoride (PVDF) membrane at 100 V for 1.5 h. After that, the PVDF membrane was blocked with 5% skimmed milk (diluted in TBS) for 1 h, washed with TBST for 10 min (3 times), and incubated with designated diluted primary antibodies at 4 °C overnight. The PVDF membrane was again washed with TBST for 10 min (3 times), then incubated with secondary antibodies at room temperature (24–26 °C) for 1 h, washed with TBST for 10 min (3 times), and finally reacted with horseradish peroxidase (HRP) solution (HRP peroxide solution:HRP substrate reagents = 1:1) (Western Lightning ECL Pro; PerkinElmer, Waltham, MA, USA) to prepared for the chemiluminescent imaging in an ImageQuant^TM^ LAS 4000 (General Electric, Boston, MA, USA). SDS-PAGE was performed in Mini-PROTEAN^®^ Tetra Vertical Electrophoresis Cell (Bio-Rad) with PowerPac^TM^ Basic Power Supply (Bio-Rad, Hercules, CA, USA). Protein expressions (CSF-1 and phospho-Ser536-p65) were measured with the monoclonal anti-CSF-1 antibody or anti-phospho-Ser536-p65, both specified in Section 2.2. Signals were standardized using the monoclonal anti-β-actin antibody specified in Section 2.2.

### 2.6. Real-Time Quantitative Polymerase Chain Reaction (Q-PCR)

RNAs in HECCs were extracted using NucleoZol (MACHEREY-NAGEL, Düren, Germany) according to the manufacturer’s instructions. The extracted RNAs were reversed-transcribed into cDNAs by iScript™ Reverse Transcription Supermix for RT-qPCR (Bio-Rad) according to the manufacturer’s protocol (PCR thermal sequences: 25 °C for 5 min, 46 °C for 20 min, 95 °C for 1 min, 4 °C for final maintenance) (T100^TM^ Thermal Cycler, Bio-Rad). Then, the cDNAs (50 ng) were mixed with primers of target genes (1 μL (10 μM) for each forward and reverse primers), 2× SYBR^®^ Green Master Mix (Bio-Rad; 10 μL), and RNAse-free water (topped to 20 μL). Finally, the mixture underwent Q-PCR thermal cycles to amplify quantitatively (95 °C for 10 min, 95 °C for 15 s, 60 °C for 30 s, plate read plus loop back to 95 °C for 15 s for 39 times, 95 °C for 10 s, 65 °C for 5 min, 65 °C to 95 °C at 0.5 °C/s (plate read)) (CFX96^TM^ Touch Real-Time PCR System, Bio-Rad). The primers used in this study were: CSF-1 (Forward) 5′-GGAGACCTCGTGCCAAATTA-3′ (Reverse) 5′-TATCTCTGAAGCGCATGGTG-3′; GAPDH (Forward) 5′-GCACAAGAGGAAGAGAGAGACC-3′ (Reverse) 5′-AGGGGAGATTCAGTGTGGTG-3′.

### 2.7. Enzyme-Linked Immunosorbent Assay (ELISA)

Sandwich ELISA (Human M-CSF/CSF1 ELISA Kit, cod. RAB0098, Sigma-Aldrich) was performed according to the manufacturer’s instructions to measure CSF-1 levels in the HECCs’ secretome. Briefly, 100 μL of HECCs’ media under each treatment group were added to appropriate wells and incubated at room temperature (24–26 °C) for 2.5 h. After that, the solution was discarded, and the wells were rinsed with 1× Wash Solution 4 times. A total of 100 μL of anti-CSF-1 antibody was added to each well and incubated at room temperature for 1 h. Then, the anti-CSF-1 antibody was discarded, while the wells were again rinsed with 1× Wash Solution 4 times, followed by adding 100 μL of secondary antibodies into each well for incubation under room temperature for 45 min. The solution was discarded while the well was rinsed 3 times with 1× Wash Solution, and 100 μL TMB One-Step Substrate Reagent was added to each well for incubation in the dark at room temperature 30 min. Finally, 50 μL of Stop Solution was added to terminate reactions in each well, after which absorbance was read at 450 nm with a spectrophotometric plate reader (SPECTROstar Nano, BMG Labtech, Ortenburg, Germany). 

### 2.8. Cell Enumeration Assay

The trypan blue dye exclusion test determined the number of cells: HEECs and HECCs were seeded in 24-well plates (2 × 10^4^ cells/well), harvested on designated days by trypsin/EDTA (200 μL) after haloperidol treatment, and enumerated with a Neubauer hemocytometer after trypan blue staining (trypan blue:cell suspension = 1:1).

### 2.9. Cell Viability Assay

Cell viability was determined using the 3-[4,5-dimethylthiazol-2-yl]-2,5-diphenyltetrazolium bromide (MTT) assay. HEECs and HECCs were seeded in 96-well plates (1 × 10^5^ cells/0.2 mL medium/well), and their media were removed at 48 h after treatments. Each well was rinsed with serum-depleted medium 2–3 times; thereafter, 200 µL of MTT (5 mg/mL) was added to each well. The cells were later cultured for 4 h in the CO_2_ incubator, after which serum-depleted medium was replaced with dimethyl sulfoxide (DMSO) to dissolve formazan, the reduced tetrazolium dye. The 96-well plates were then centrifuged (3500 rpm, 5 min), and 50 μL of the purpled solution in each well was pipetted before being transferred into fresh 96-well plates. Absorbance was read at 570 nm using a spectrophotometric plate reader (SPECTROstar Nano, BMG Labtech, Ortenberg, Germany).

### 2.10. Cell Migration and Invasion Assays

Cell migrations for HECCs were observed in the 24-well Transwell^®^ (Corning^®^ Inc., Corning, CA, USA). In principle, Transwell^®^ was first tended for equilibrium by adding HECCs’ medium with 10% FBS in the lower compartment and medium without 10% FBS in the upper compartment for 1 h. Then, HECCs (2 × 10^5^ per well), with accompanied treatment and FBS-depleted medium, were added into the upper compartment for the indicated incubating periods. After treatment and incubation periods, the FBS-depleted medium was syringed out from the upper compartment, and HECCs were fixed by submerging the cells with 100% methanol and gently wiping the upper surface of the Transwell^®^ membrane with cotton buds. The migrating capabilities of HECCs were examined and compared by enumerating the presence of cells on the lower side of the Transwell^®^ membrane using light microscopy (4 fields per well). Cell invasions were also assessed by the 24-well Transwell^®^ but with pre-coated Matrigel^TM^ on its membrane (2.5 mg/mL; Becton Dickinson, Franklin Lakes, NJ, USA). The protocol for cell invasions was the same as that for cell migrations, with the exception of different treatment conditions. 

### 2.11. Cell Cycle Assay

HEC1A was harvested with trypsin and later centrifugated under 1500 rpm for 5 min. The resulting pellets were fixed in cold 70% ethanol, rinsed with PBS, and resuspended in 0.5 mL of PBS mixed with propidium iodide (PI, 500 μg/mL) and RNase A (1 mg/mL). After incubation in the dark in the CO_2_ incubator for 30 min, the cells were centrifugated, and their supernatant was replaced with PBS. The cells were analyzed by a flow cytometer in the PI/RNase A solution (same concentration as previously mentioned) (FACSCanto^TM^ and CellQuest^TM^ software; Becton Dickinson, Franklin Lakes, NJ, USA).

### 2.12. Cell Proliferation Assay

The nonisotopic immunoassay, Calbiochem^®^ (Sigma-Aldrich), was used to quantify cell proliferation in HEECs and HECCs by detecting the incorporation of immunolabeled BrdU into newly synthesized DNA. The steps are performed under the manufacturer’s instructions. Concisely, 1.5 × 10^5^ cells in 100 μL of appropriate media were seeded into each well of a 96-well plate. Then cells underwent different treatments before 20 μL of diluted BrdU solution was added to each well and incubated for 24 h in the CO_2_ incubator. After the solution was discarded, 200 μL of the Fixative/Denaturing Solution was added to each well, and the cells were incubated at room temperature (24–26 °C) for 30 min. The solution was again discarded, after which 100 μL of diluted anti-BrdU antibody was added to each well, and the cells were incubated for 1 h at room temperature. Each well underwent rinsing with 1× Wash Buffer 3 times. Peroxidase Goat Anti-Mouse IgG HRP Conjugate was diluted, 100 μL of which was added to each well with incubation at room temperature for 30 min. Wells were rinsed with 1× Wash Buffer 3 times, incubated with 100 μL Substrate Solution in the dark for 15 min, and finally, reacted with Stop Solution in the same manner as Substate Solution. Absorbances were measured at dual wavelengths (450, 540 nm) immediately after the final step in a spectrophotometric plate reader (SPECTROstar Nano, BMG Labtech, Ortenberg, Germany).

### 2.13. Soft Agar Colony Formation Assay

The soft agar colony formation assay was utilized to monitor the anchorage-independent transformation of HEECs after haloperidol treatment. Steps were performed according to the published protocol [23]. Briefly, 5 × 10^3^ HEECs were mixed with 0.3% agarose containing 0, 10, 50, or 100 μM haloperidol and were plated over a 0.6% agarose layer pre-mixed with the medium. The medium was renewed twice a week for the upper agarose layer. After 28 days, the cells were cross-linked with paraformaldehyde (4%), rinsed with PBS, and stained with crystal violet solution (0.05%). The colonies were imaged by a camera-mounted inverted microscope at 40× magnification, and the number of colonies was subsequently enumerated.

### 2.14. Gene Expression Profiling Assay

RT^2^ Profiler PCR Array^TM^ Human Signal Transduction PathwayFinder™ (330231/PAHS-014Z; Qiagen, Hilden, Germany), a pathway-focused Q-PCR assay, was utilized to screen 84 genes indicative of 10 different signal transduction cascades (TGF-β, WNT, NF-κB, JAK/STAT, p53, Notch, Hedgehog, PPAR, Oxidative Stress, Hypoxia Signalling) in HEC1A after haloperidol treatments. Steps were conducted according to the manufacturer’s instructions. In principle, RNAs were extracted from HEC1A with or without haloperidol treatment and reversed-transcribed to cDNAs following the Q-PCR procedures in Section 2.6. Then, cDNAs (50 ng/μL, 102 μL) were mixed with 2× RT^2^ SYBR^®^ Green qPCR Mastermix (1350 μL) (Qiagen) and RNAse-free water (1248 μL), after which 25 μL of the solution was pipetted into each well in a plate of Q-PCR array. The cDNAs underwent thermal cycles to be amplified quantitatively (95 °C for 10 min, 95 °C for 15 s, 60 °C for 1 min, loop back to 95 °C for 15 s for 39 times, 95 °C for 1 min, 65 °C for 2 min (camera off), 65 °C to 95 °C at 2 °C/min (camera on)) (CFX96^TM^ Touch Real-Time PCR System, Bio-Rad). The resulting data were analyzed with tools on the GeneGlobe Data Analysis Center (Qiagen).

### 2.15. Animal Studies

To characterize how chronic haloperidol treatment influences the status of endometrial proliferation in vivo, six-week-old female C57BL/6J mice were subjected to ovariectomy. In these mice, oviducts were ligated, and their ovaries were removed under isoflurane anesthesia according to the published protocol [24]. A total of 2 weeks after ovariectomy, mice received vehicle (normal saline), haloperidol (2 mg/kg/day, Sigma-Aldrich), haloperidol plus vehicle (20% Captisol^®^), or haloperidol plus BLZ945 (200 mg/kg/day, Chemshuttle) by orogastric administration for 28 days. To determine whether chronic haloperidol treatment promotes endometrial cancer progression, eight-week-old female NOD SCID mice were established as the orthotopic xenograft model of human EC following the method published in [25]. A total of 1 × 10^6^ of HEC1A-lucGFP resuspended in 50 μL of Matrigel^®^ were implanted onto the endometrium by the uterine puncture. After tumor inoculation, mice were treated with similar prescriptions to the six-week-old female C57BL/6J mice for 21 days. In the pharmacokinetic studies, haloperidol (1 mg/kg) was administered intravenously to the eight-week-old female C57BL/6J mice. Blood samples with the corresponding uteri and brains were collected at 0.5, 1, 3, 6, 12, and 24 h after drug administration. Plasma was separated from the tissues by centrifugation (5000 rpm; 10 min; 4 °C). Separated plasma and tissues were stored at −70 °C until analysis.

### 2.16. Immunohistochemistry

Immunohistochemical staining was performed on uterine tissues by the avidin-biotin-peroxidase complex (ABC) method. The uterine tissues were cross-linked with formalin, embedded in paraffin, sectioned, and mounted on microscopic slides. The sections were dewaxed and underwent rehydration in a gradient of ethanol solutions (50 to 100%). Endogenous peroxidases were quenched with H_2_O_2_ incubation (0.3%; 15 min). The sections were incubated with Ki-67 primary antibody (diluted 1:100; cod. ab21700; abcam, Cambridge, UK) overnight at 4 °C, after which conjugated to the fluorophore-coated secondary antibody and counterstained with Delafield’s hematoxylin. The sections were finalized with dehydration and mounting. Negative controls were stained without primary antibodies. The number of positive endometrial nuclei under Ki-67 was enumerated and expressed as percentage values (per 1000 nuclei).

### 2.17. Bioluminescent Imaging (BLI)

Mice bearing HEC1A-lucGFP were imaged by the IVIS Imaging System 200 Series (PerkinElmer) to monitor the bioluminescence from the inoculated tumors. Mice were first anesthetized with isoflurane (2.5%) in the imaging chamber and intraperitoneally injected with D-luciferin (250 µg/g body weight; PerkinElmer). BLI was captured 15 min after D-luciferin injection. For quantitative analyses of the BLI signal, signaling intensities in the acquired intrauterine region were defined as the total number of photons (s^−1^; steradian^−1^; centimeter^−2^) with Living Image Software (ver. 2.60.1).

### 2.18. LC-MS Analysis

Haloperidol was extracted from the plasma, brain, and uterine tissues and prepared for liquid chromatography-mass spectrometry (LC-MS) (LCMS-2020, Shimadzu Corp., Kyoto, Japan) analyses according to the following articles [26,27]. Briefly, tissues were homogenized in methanol containing 0.2 mL of ice-cold KCl solution (1.15%; mass concentration) and 0.3 mL of acetic acid (2%; volume concentration). For the preparation of plasma extracts, 20 μL of it was mixed into 130 μL of hexane plus dichloromethane (70:30, in volume). After the mixture was vortexed and settled, 100 μL of the organic layer was relocated into an evaporation tube. The solution was desiccated by SpeedVac^TM^ (Thermo Fisher Scientific), and the residue was restored into 20 μL of acetonitrile water (50:50 in volume). To prepare both brain and uterine extracts, they were dissolved with acetonitrile water (50:50 in volume; brain extracts: 5 mL; uterine extracts: 3 mL), centrifuged with 5200× *g* for 10 min at 25 °C, and the corresponding supernatants were each transferred to a new 15 mL tube. A total of 3 mL of hexane plus dichloromethane (70:30, in volume) was added to 500 μL of the supernatant. After the mixture was vortexed and settled, 2 mL of the organic layer was relocated into an evaporation tube. The solution was desiccated by SpeedVac^TM^. After desiccation, the residue was restored in 500 μL of acetonitrile water (50:50 in volume). For the proper LC-MS analyses, 2 μL of each sample was injected into the C18 column (2.6 μm, 2.1 i.d. × 100 mm., SunShell, ChromaNik, Osaka, Japan). The mobile phase was constituted of 2 mM ammonia acetate with 0.1% formic acid in ddH2O. Haloperidol ion was detected in positive mode at *m*/*z* value 376.2+.

### 2.19. The Nationwide Population-Based Cohort Study

The epidemiological study investigating the association between the use of haloperidol and EC survival was carried out by analyzing data sets in the National Health Insurance Research Database (NHIRD) of Taiwan [28]. Patients newly diagnosed with EC (ICD-9-CM 182, conforming with the International Classification of Disease, 9th Revision, Clinical Modification) from 2003 to 2011 in the NHIRD were enrolled. Additionally, patients with previous cancer were excluded. Every prescription of haloperidol on or after the diagnosis of each EC patient was collectively designated as a single use of haloperidol in the haloperidol treatment group. The Cox proportional hazards model was utilized to estimate the EC-specific hazard ratios (HRs) and 95% CIs between the use of haloperidol and overall survival after adjusting for other variables. The Kaplan–Meier methodology and the log-rank test were implemented to inspect the survival discrepancy between users and non-users. Statistical significance was established on *p*-value < 0.05 or 95% CI. Statistical analyses were undertaken with SAS (SAS Institute, ver. 9.4, Cary, NC, USA).

### 2.20. Statistical Analyses

All data were presented as mean  ±  SD using one-way ANOVA with post-hoc Scheffé’s adjustment. The significance of differences between the means in control and experimental groups was assessed with the unpaired two-sample Student’s *t*-test, where statistical significance was *p*-value < 0.05.

## 3. Results

### 3.1. Haloperidol Promotes Endometrial Hyperplasia and Malignant Progression of Endometrial Cancer (EC) In Vivo

The median daily dose of haloperidol in clinical practice is about 10 mg [29]. Hence, we performed the dose conversion between humans and mice with a virtual dosage converter, DoseCal [30], and calculated that the mouse equivalent dose for a human weighing 60 kg is about 2 mg/kg/day. Based on this rationale, we first observed the effect of chronically treating haloperidol (2 mg/kg/day for 28 days) on the proliferation status of endometrial cells in vivo using Ki-67 immunostaining. To avoid hormone-induced cell proliferation resulting from estrous cycles, we measured Ki-67 expression in mice with ovariectomy and at the diestrous stage. Low levels of Ki-67 staining were found in the endometrium of control mice (Figure 1A,B). Alternatively, high levels of Ki-67 staining were observed in haloperidol-treated mice, indicating that chronic treatment of haloperidol induces hyper-proliferation in the endometrium of mice. To further understand the effect of chronically treating haloperidol on EC progression in vivo, we established an orthotopic xenograft mice model of HEC1A-lucGFP and monitored its tumor growth by BLI. After being exposed to haloperidol for 21 days upon the implantation of carcinoma cells (Figure 1C), these mice expressed enhanced tumor growth and metastasis (Figure 1D–F). Together, these results indicated that chronically treating haloperidol in clinically relevant doses promotes endometrial hyperplasia and malignant progression of EC in vivo.

### 3.2. Haloperidol Highly Accumulates in the Uterus

We next investigated the pharmacokinetics of haloperidol in mice’s plasma, brain, and uterine tissues. By analyzing the haloperidol concentrations with LS-MS after its intravenous administration at 1 mg/kg, we discovered that the peak concentrations of haloperidol in all three tissues were at 0.5 h after administrating haloperidol (Figure 2A). Haloperidol had a uterus-to-plasma concentration ratio between 18 and 62 and a brain-to-plasma ratio between 7 and 21 within 24 h after intravenous injection (Figure 2B). A total of 12 h after administrating haloperidol, the levels of haloperidol in the uterus were 62 times higher than those in plasma. In contrast, the brain-to-plasma concentration ratio was only 21. This ratio was similar to those derived from previous studies [31]. These results showed that haloperidol can highly accumulate in the uterus.

### 3.3. Haloperidol Promotes the Cellular Transformation of Human Endometrial Carcinoma Cells (HECCS) and the Malignant Progression of HECCS

As to further examine the effect of haloperidol on HEECs and HECCs, we treated HEECs with haloperidol at various concentrations for 28 days in the soft agar colony formation assay while also exposing HECCs to the medium with different concentrations of haloperidol for 24 h and measured their tumor-cell viability, migration, and invasion. Under these conditions, HEECs formed colonies in a dose-dependent relationship to haloperidol (Figure 3A,B) and presented dramatic changes in their cellular morphology (Appendix A). On the other hand, HECCs thrived accordingly in a dose-dependent manner with haloperidol (Figure 3C), multiplied significantly as indicated by an upscaled G2/M population as well as nuclear incorporation (Figure 3D–F), and underwent enhanced migration and invasion (Figure 3G,H). These data specified that haloperidol can promote cellular proliferation in HEECs and HECCs, induce cellular transformation in HEECs, and stimulate malignant progression in HECCs.

### 3.4. Haloperidol Induces CSF-1 Production and Secretion in HECCs via NF-κB Activation

For a clearer mechanistic insight on the haloperidol-mediated cell growth and malignancy of HECCs, we then applied RT^2^ Profiler PCR array^TM^, a human signal transduction pathway finder comprising 84 key genes responsive to activation or inhibition of signal transduction pathway, to HEC1A cells with or without haloperidol exposure. Haloperidol treatment in HEC1A cells increased the expression of target genes involved in TGF-β, WNT, and NF-κB pathways, while also decreasing the expression of target genes in the NF-κB, WNT, and Hedgehog pathways (data not shown). Among these genes, expression changes beyond 2-fold were found in *TNFSF10*, *DAB2*, *CSF-1*, *TNF*, *BMP2*, *BMP4*, *WNT2B*, and *WNT6* (Figure 4A). Interestingly, CSF-1 is a growth factor associated with the histopathological grades of the endometrial carcinoma [32] and plays a role in carcinogenesis, tumorigenesis, and metastasis in other cancers [33,34]. Therefore, we verified the expression of CSF-1 in HECCs with Q-PCR, WB, and ELISA assays, confirming that haloperidol treatment significantly increased the production and secretion of CSF-1 (Figure 4B–F). Furthermore, the treatment of haloperidol promoted the activation and signaling activities of NF-κB, an upstream regulator of CSF-1, in HECCs (Figure 4G–I). Blockade of NF-κB using its inhibitors, SN50 and BAY 11-7082, repressed the haloperidol-induced production and secretion of CSF-1 (Figure 4J,K). As a result, these outcomes suggested that haloperidol induces the production and secretion of CSF-1 in HECCs via NF-κB activation.

### 3.5. CSF-1/CSF-1R Stimulates the Cellular Transformation of HEECs and the Malignant Progression of HECCs In Vitro

The roles of CSF-1 and its receptor, CSF-1R, in HEECs and HECCs are unknown. Therefore, we characterized the role of CSF-1 in the cellular transformation of HEECs and the malignant progression of HECCs. Supplementation with CSF-1 recombinant protein (rCSF-1) in HEECs induced the formation of colonies in soft agar assay (Figure 5A); it also significantly enhanced the cell viability, migration, and invasion of HECCs compared to untreated cells (Figure 5B–D). Based on the above results, we hypothesized that haloperidol might induce the cellular transformation of HEECs and the malignant progression of HECCs via the CSF-1/CSF-1R axis. To test this hypothesis, HEECs and HECCs were treated with CSF-1-specific neutralizing antibody (CSF-1 n.b.) after being treated with haloperidol or vehicle (normal saline). Neutralization of CSF-1 significantly inhibited the colony formations and cell viability of HEECs, while also inhibiting cell viability, migration, and invasion of HECCs. At the same time, similar effects of CSF-1 n.b. could also be observed in the haloperidol-treated groups (Figure 5A,E–G). Concordantly, HEECs and HECCs treated with BLZ945, an inhibitor of the colony-stimulating factor-1 receptor (CSF-1R), after exposure to haloperidol possessed comparable effects to those of CSF-1 n.b. (Appendix A). These findings pointed out that the CSF-1/CSF-1R axis can promote carcinogenesis for HEECs and malignancy for HECCs in vitro.

### 3.6. Blockage of CSF-1/CSF-1R Inhibits Haloperidol-Promoted Endometrial Hyperplasia and Malignant Progression of EC In Vivo

To explore the role of the CSF-1/CSF-1R axis in haloperidol-induced endometrial hyperplasia in vivo, we ovariectomized the mice and fed them haloperidol (2 mg/kg/day) plus BLZ945 (200 mg/kg/day) or haloperidol plus vehicle (normal saline) by orogastric administration for 28 days. Immunohistochemical assay of Ki-67 in uterine tissues showed that compared with the vehicle group, levels of Ki-67 staining were lesser in the BLZ945 group, suggesting that BLZ945 can halt haloperidol-induced endometrial hyperplasia (Figure 6A,B). Next, we observed the effects of BLZ945 in halting the progression of EC. Mice transplanted with HEC1A-lucGFP and given BLZ945 displayed significant inhibition of tumor growth and metastasis compared to those receiving vehicles only (Figure 6C–E). Therefore, blockage of endogenous CSF-1/CSF-1R signaling can inhibit EC’s tumor growth and malignant progression. Furthermore, mice administrated with haloperidol plus BLZ945 also showed significant inhibition of haloperidol-induced tumor growth and metastasis than those receiving haloperidol plus vehicle (Figure 6C–E). These results revealed that the CSF-1/CSF-1R axis promotes haloperidol-induced endometrial hyperplasia and participates in the proliferation and malignant transformation of EC, regardless of receiving haloperidol.

### 3.7. Using Haloperidol Reduces the Survival Outcome of EC Patients

Finally, we investigated the relationship between survival outcomes and the use of haloperidol in EC patients through a nationwide cohort study in Taiwan. A total of 9502 patients with EC were included in our study from 2003 to 2011. A total of 80 patients were enrolled in the use of haloperidol, and 9422 patients were enrolled in the non-statin cohorts. The risk of death in EC patients using haloperidol was higher than that in EC patients without using haloperidol, with the crude OR (95% CIs) at 1.94 (1.25–3.02) (*p*-value < 0.01) and the multivariate-adjusted ORs (95% CIs) at 1.75 (1.12–2.72) in model 1 (*p*-value < 0.05) and 1.46 (1.27–2.44) in model 2 (*p*-value > 0.05) (Table 1). The Kaplan–Meier curve revealed that the survival probability in EC patients using haloperidol was lower than that in EC patients without using haloperidol (*p*-value = 0.003) (Figure 7). These results suggested that the use of haloperidol during EC progression decreases the survival probability in EC patients.

## 4. Discussion

Prolonged treatment with antipsychotics may appear in patients diagnosed with mental disorders or cancers. Still, their long-term side effects, including the impacts on tumor initiation, promotion, and progression, are challenging to evaluate and remain largely unknown [35]. We showed for the first time that haloperidol, prophylaxis frequently used for delirium in patients with advanced cancer [3], induces hyper-proliferation in the endometrium of mice and promotes tumor progression in the orthotopic xenograft model of HEECs under clinically relevant doses. Moreover, in our population-based cohort study on EC patients, the prescription of haloperidol was related to an upscaled mortality rate. Our cell and animal studies further unveiled that haloperidol promotes EC progression by directly accumulating in the uterus, activating NF-κB, and stimulating its downstream signaling target, CSF-1/CSF-1R. These findings suggested that haloperidol is potentially detrimental for women and should be considered a risk factor for patients with EC. 

Several mechanisms have been hypothesized to justify the carcinogenic capabilities of antipsychotics. There is increasing evidence that elevated prolactin, via the hypothalamic dopamine system and perhaps pituitary dopamine receptors, could be the instigator for some antipsychotics-induced EC [36,37]. Nevertheless, the epidemiological relationship between antipsychotics-mediated hyperprolactinemia and EC is limited and conflicting [20,38]. A case-control study revealed that using antipsychotics is a risk factor for EC, while antipsychotic-linked hyperprolactinemia could induce this EC propensity [20]; however, a recent cohort study reported a contradictory finding [38]. Therefore, the linkage between antipsychotics-mediated hyperprolactinemia and EC is insufficient to acknowledge antipsychotics as a risk determinant for EC. Alternatively, multiple antipsychotics have been revealed to be genotoxic and perhaps carcinogenic across different cancer models; yet, haloperidol was tested negatively in genotoxicity assays but positively in long-term carcinogenesis assays [39], suggesting that haloperidol-induced carcinogenesis and the subsequent progression are not the results of genetic mutation incurred through DNA damage. Indeed, we demonstrated that the dysregulated autocrine signaling axis, NF-κB/CSF-1/CSF-1R, underlies this haloperidol-mediated carcinogenesis in the orthotopic xenograft model of HEECs. This finding provides the fundamental reasoning behind the haloperidol-linked EC risk and elucidates a novel mechanism for antipsychotic-induced carcinogenesis.

If haloperidol is a potential compound for the carcinogenesis of EC, it questions whether EC patients concurrently intaking haloperidol can have a worsened prognosis. Our nationwide population-based cohort study disclosed that using haloperidol in EC patients during EC progression reduces their survival probability. Recently, another nationwide population-based cohort study reported that exposure to haloperidol is associated with a higher mortality rate compared to other antipsychotics [40]; supplementary to this finding, our epidemiological study suggests that haloperidol-related poor outcomes in EC patients might contribute to this higher risk of mortality. Moreover, our in vitro and in vivo studies confirmed that haloperidol can promote the malignant progression of EC, further indicating that haloperidol is a risk compound for the prognosis of EC. These findings provide a warning message that the use of haloperidol during EC progression might increase mortality by enhancing tumor malignancy. Future investigations are subsequently necessary to delineate the contribution of haloperidol to the risk of other types of cancer.

We discovered that multiple genes were upregulated and downregulated in EC cells by haloperidol treatment. Among the 10 signaling pathways that were significantly altered, the most notable one was the upregulation of the well-characterized NF-κB signaling cascade. Aberrant activation of the NF-κB signaling cascade, either through genetic or epigenetic alterations, has been proven to promote carcinogenesis, tumorigenesis, and metastasis in many cancer [41]. Additionally, dysfunction of NF-κB signaling is also implicated in endometrial carcinogenesis and EC progression [42]. Interestingly, haloperidol also induces the activation of NF-κB in clonal hippocampal cells, imparting oxidative toxicity to these neuronal cells [43]. Although it remains elusive by which mechanism haloperidol activates the NF-κB cascade, our studies suggest that the NF-κB signaling cascade participates in haloperidol-mediated initiation, progression, and metastasis of EC.

CSF-1 is a hematopoietic growth factor that maintains the normal physiological functions, including the survival, proliferation, differentiation, and mobilization, of cells from the myeloid pedigree [34]. Apart from its regular physiological functions, it also plays a role in carcinogenesis and tumor progression. Mouse NIH 3T3 cells that overexpressed the human protooncogene, *c-FMS* (encoding human CSF-1R), underwent rapid cellular growth after replacing recombinant CSF-1 into their culturing medium [44]; this indicates that the gain-of-functions of CSF-1 and its receptor induces cellular transformation. Moreover, CSF-1/CSF-1R is heavily involved in mammary gland development and carcinogenesis of breasts and ovaries [32]. Transgenic mice overexpressing CSF-1 in their mammary epithelium presented an increased breast cancer metastatic potential [45], while overexpression of CSF-1/CSF-1R in human ovarian and breast carcinoma correlates with poor survival outcomes [32]. In the case of EC, CSF-1 and its receptor have been revealed to overexpress in human endometrial adenocarcinoma, and the serum CSF-1 levels in patients with EC were significantly elevated in active or recurrent EC patients [46]. The present study provides direct evidence showing CSF-1 promotes proliferation, migration, and invasion of HECCs. Most importantly, our data unveiled that haloperidol can induce CSF-1 production and secretion in HECCs, contributing to carcinogenesis and malignant progression in vitro and in vivo. Therefore, blockage of CSF-1 or its receptor-ligand interaction provides a strategy to inhibit endogenous or exogenous CSF1-promoted tumorigenesis and progression.

## 5. Conclusions

In summary, our in vitro and in vivo studies demonstrated that haloperidol is a carcinogenic compound for EC and promotes the malignant progression of EC cells, while our epidemiological study consolidated that the use of haloperidol is unfavorable to the survival outcome of EC patients. We also provided unique insight into haloperidol-promoted cell proliferation and tumor progression: haloperidol upregulates the NF-κB/CSF-1/CSF-1R signaling and engenders EC’s malignant transformation and progression; thereby, in vitro and in vivo blockage of CSF-1 or its receptor-ligand interaction reverses these haloperidol-promoted effects on EC. Physicians should be cautious about prescribing haloperidol in female patients, especially with EC.

## Figures and Tables

**Figure 1 cancers-14-03089-f001:**
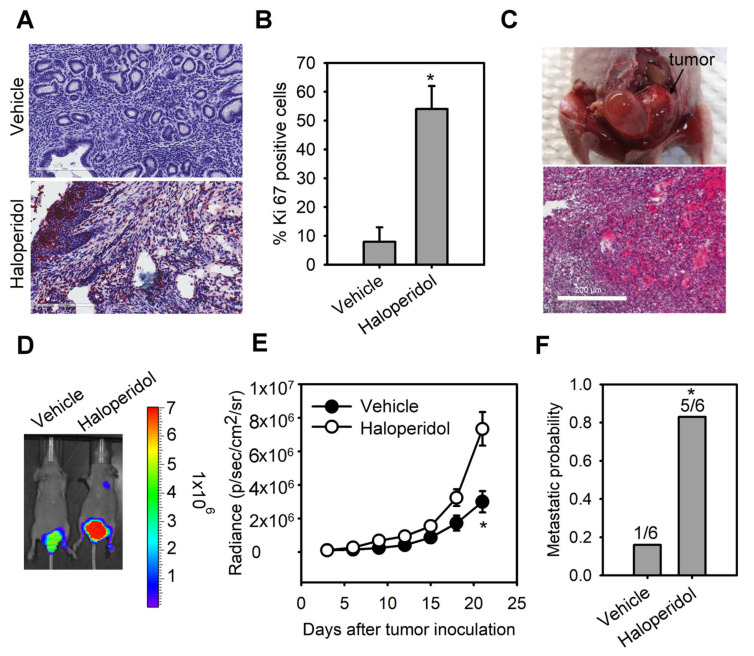
Haloperidol promotes endometrial hyperplasia and malignant progression of endometrial cancer (EC) in vivo. Ki-67 immunohistochemistry (**A**) and its quantification (**B**) in uterine tissues from adult C57BL/6J mice receiving the chronic treatment of haloperidol (2 mg/kg/day for 28 days). Bar = 200 μm. (**C**) Representative images of an orthotopic xenograft model of HEC1A-lucGFP developed in the uterus from athymic nude mice and its H&E staining. Bar = 500 μm. (**D**) Bioluminescent images of HEC1A-lucGFP orthotopic xenografts with or without the chronic treatment of haloperidol (2 mg/kg/day for 21 days) on day 21 after tumor inoculation. (**E**) Longitudinal monitoring of the tumor growth via BLI signal intensities for each treatment group in (**D**). (**F**) The metastatic probability for each treatment group in (**D**). Data are presented as means ± SD (n = 6). * *p* < 0.01 compared to the untreated or vehicle group (normal saline).

**Figure 2 cancers-14-03089-f002:**
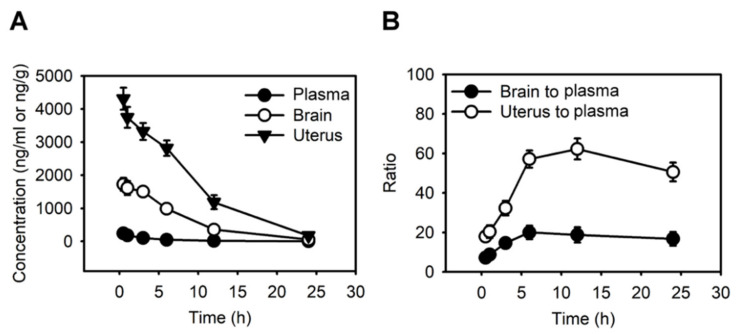
Haloperidol highly accumulates in the uterus. (**A**) The concentration-time profiles of haloperidol in plasma, brain, and uterus of adult C57BL/6J mice following single intravenous administrations of 1 mg/kg haloperidol. (**B**) Brain-to-plasma and uterus-to-plasma concentration ratios for haloperidol. Data are presented as means ± SD (n = 6).

**Figure 3 cancers-14-03089-f003:**
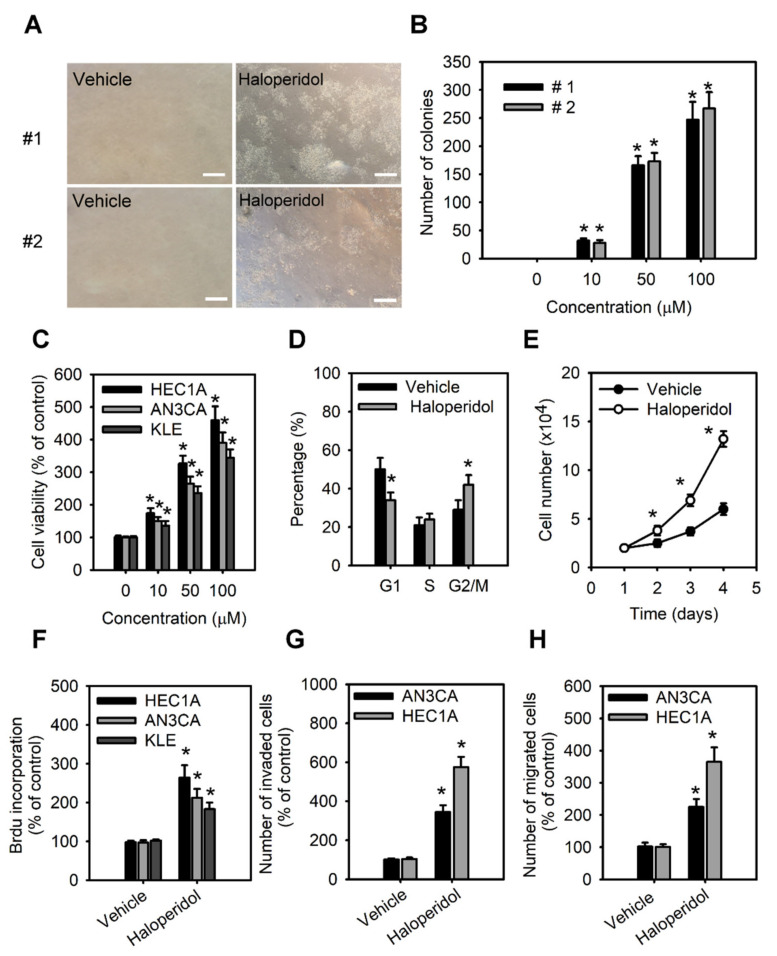
Haloperidol promotes the cellular transformation of human endometrial carcinoma cells (HECCs) and the malignant progression of HECCs. (**A**) Representative micrographs of colonies in soft agar colony formation assay of primary HEECs treated with haloperidol (100 μM) for 28 days. (**B**) The number of colonies formed in soft agar for primary HEECs treated with 2 weeks of haloperidol (0–100 μM). (**C**) Cell viability was increased in a dose-dependent manner by 0, 10, 50, and 100 μM of haloperidol in HEC1A, AN3CA, and KLE cells for 48 h. (**D**) Cell cycle distribution of HEC1A cells receiving 10 μM of haloperidol for 48 h. (**E**) Cell growth curve of HEC1A cells with or without 10 μM of haloperidol in the culture medium. (**F**) The nuclear incorporation of bromodeoxyuridine (BrdU) in HEC1A, AN3CA, and KLE cells. Each received 10 μM of haloperidol for 48 h. The invasion (**G**) and migration (**H**) of HEC1A and AN3CA cells. Each received 10 μM of haloperidol for 48 h. Data are presented with means ± SD within triplicate experiments. * *p* < 0.01, compared with the untreated or vehicle group (normal saline).

**Figure 4 cancers-14-03089-f004:**
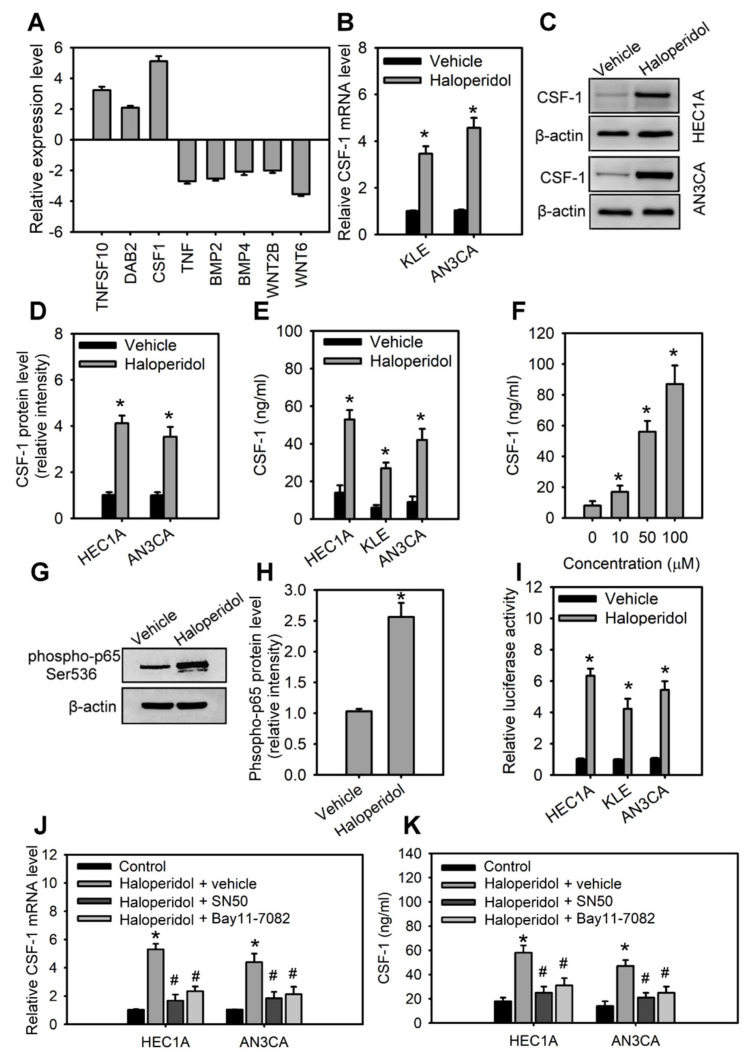
Haloperidol induces the production and secretion of CSF-1 in HECCs via NF-κB activation. (**A**) The differentially expressed genes in HEC1A with at least a 2-fold change after 48 h treatment of 10 μM haloperidol. The levels of expressions are measured with the RT^2^ Profile PCR Array^TM^. The relative mRNA levels (**B**), protein levels (**C**), relative protein densities (**D**), and concentrations of CSF-1 in the medium (**E**) of HEC1A, AN3CA, and KLE cells receiving 10 μM of haloperidol for 48 h. (**F**) Concentrations of secreted CSF-1 in the medium of HEC1A cells responded to 0, 10, 50, and 100 μM of haloperidol for 48 h. (**G**,**H**) Western blot analysis and quantification for p65 phosphorylation in HEC1A cells treated with 10 μM of haloperidol for 30 min. (**I**) NF-κB signaling activities in HEC1A, AN3CA, and KLE cells with or without the treatment with 10 μM of haloperidol for 48 h. The NF-κB signaling activities were measured with luciferase reporter assays. The relative mRNA levels (**J**) and concentrations of CSF-1 (in the conditioned medium) (**K**) of HEC1A and AN3CA cells treated for 48 h with 10 μM of haloperidol with or without the NF-κB inhibitors (SN50, 50 μg/mL; BAY 11-7082, 2.5 μM). Data are presented as means ± SD within triplicate experiments. * *p* < 0.01, compared with the untreated or vehicle group (normal saline). # *p* < 0.01, compared with the haloperidol plus vehicle group.

**Figure 5 cancers-14-03089-f005:**
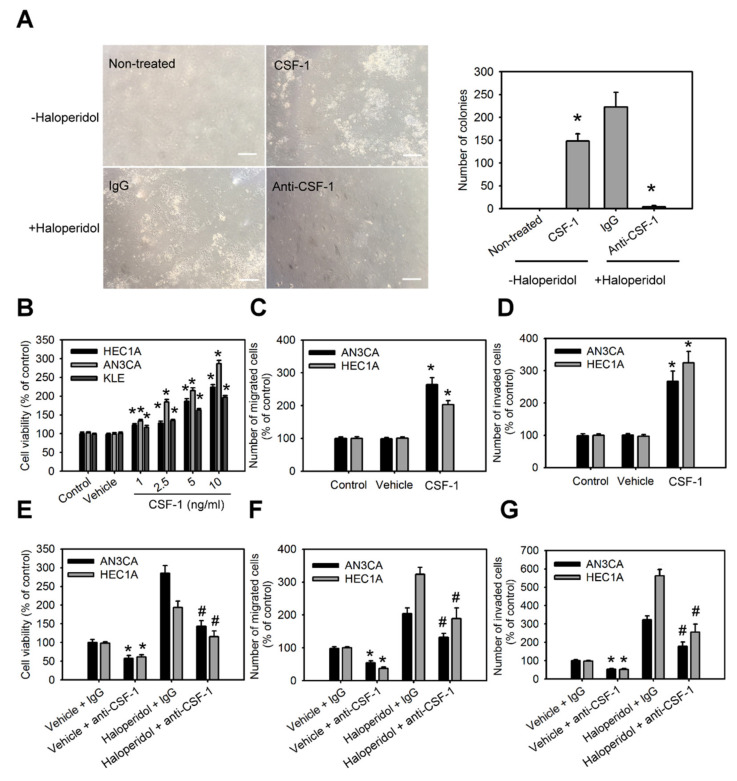
CSF-1/CSF-1R stimulates the cellular transformation of HEECs and the malignant progression of HECCs in vitro. (**A**) The number of colonies formed in soft agar of primary HEECs treated with nothing (control), CSF-1 recombinant protein (10 ng/mL), haloperidol (100 μM) plus IgG, or haloperidol plus CSF-1 neutralizing antibody (100 ng/mL) for 28 days. Cell viability (**B**), migration (**C**), and invasion (**D**) of HEC1A, AN3CA, and KLE cells treated with nothing (control), vehicle, or CSF-1 recombinant protein (10 ng/mL) for 48 h. Cell viability (**E**), migration (**F**), and invasion (**G**) of HEC1A and AN3CA cells treated with vehicle plus IgG, vehicle plus CSF-1 neutralizing antibody (100 ng/mL), haloperidol (100 μM) plus IgG, or haloperidol plus CSF-1 neutralizing antibody (100 ng/mL) for 48 h. Data are presented as means ±SD within triplicate experiments. * *p* < 0.01, compared with the untreated or vehicle groups (normal saline). # *p* < 0.001, compared with the controlled IgG group.

**Figure 6 cancers-14-03089-f006:**
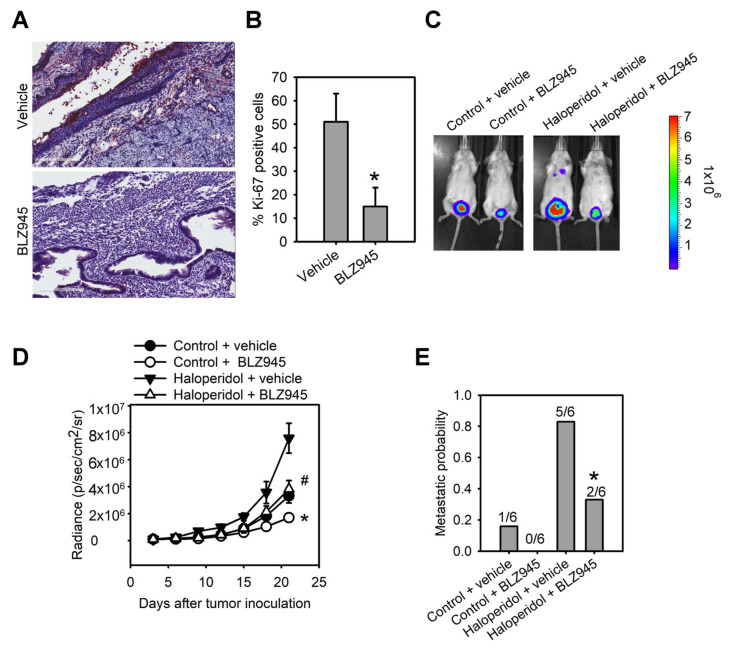
Blockage of CSF-1/CSF-1R inhibits haloperidol-promoted endometrial hyperplasia and malignant progression of EC in vivo. Ki-67 immunohistochemistry (**A**) and its quantification (**B**) in uterine tissues from adult C57BL/6J mice receiving chronic treatment of vehicle or haloperidol (2 mg/kg/day) plus BLZ945 (200 mg/kg/day) for 28 days. Bar = 200 μm. (**C**) Bioluminescent images of the HEC1A-lucGFP bearing mice, which either received treatments of vehicle, BLZ945 (200 mg/kg/day for 21 days), haloperidol (2 mg/kg/day for 21 days) plus vehicle, or haloperidol (2 mg/kg/day for 21 days) plus BLZ945 (200 mg/kg/day for 21 days) on day 21 after tumor implantation. (**D**) Longitudinal monitoring of the tumor growth via BLI signal intensities for each group of treatments in (**C**). (**E**) The metastatic probabilities for each group of treatments in (**C**). Data are presented as means ± SD (n = 6). * *p* < 0.01 compared to the vehicle’s group (normal saline). # *p* < 0.001, compared with the vehicle’s group (normal saline).

**Figure 7 cancers-14-03089-f007:**
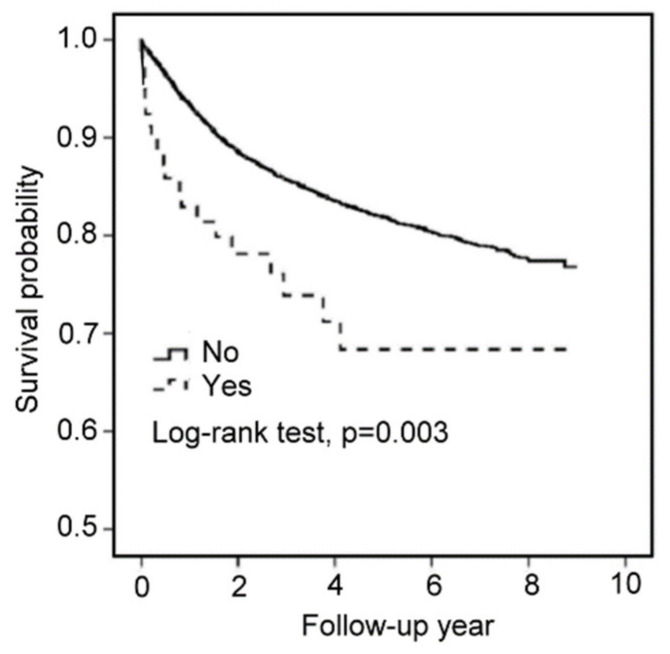
*Using haloperidol reduces the survival outcome of EC patients*. Kaplan–Meier survival curves for EC patients with or without haloperidol.

**Table 1 cancers-14-03089-t001:** Proportion of death among medicine use in endometrial carcinoma (N = 9502).

Haloperidol					HR (95% CI)		
	N	Death No	Person-Years	Rate ^a^	Crude	Model 1 ^b^	Model 2 ^c^
No	9422	1351	31,754	42.55	1.00	1.00	1.00
Yes	80	20	241	82.94	1.94 (1.25–3.02) **	1.75 (1.12–2.72) *	1.46 (0.87–2.44)

^a^ Rate, per 1000 person-years, ^b^ Model 1, adjusted for age, diabetes, hypertension, hyperlipidemia, and polycystic ovaries, ^c^ Model 2, added adjustment of other antipsychotics in model 1, * *p*-value < 0.05; ** *p*-value < 0.01.

## Data Availability

The data presented in this study are all available within the article.

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
