# Peer review of "Haloperidol Instigates Endometrial Carcinogenesis and Cancer Progression by the NF-κB/CSF-1 Signaling Cascade"

_cancers, 2022, doi:10.3390/cancers14133089_

Round 1

Reviewer 1 Report

The objective of the authors was to investigate the effect of Haloperidol usage (an antipsychotic drug) on development of endometrial cancer (EC). The authors have also investigated the association between the use of halperidol treatment and EC-survival rate using population-based cohort study with datasets from National Health Insurance Research Database (NHIRD) of Taiwan. The authors have reported that chronic treatment of haloperidol induced hyper-proliferation in the endometrium of mice and promotes endometrial hyperplasia and malignant progression of EC in orthotopic xenograft mice model. The authors have also reported that haloperidol promotes EC progression in vitro and in vivo through nuclear factor kappa B (NF-κB) mediated activation of CSF-1. The authors have also reported use of haloperidol decreases the survival probability in EC patients. It is interesting data but the manuscript is poorly written and could not be considered for publication in the current form as authors are requested to address several concerns as listed in comments for authors section.

Major concerns:

1) Could authors please consider to elaborate the introduction especially on studies implicating antipsychotics as a risk factor for endometrial cancer. The authors have merely mentioned it and brief elaboration on this topic is required for the introduction.

2) Could authors please rewrite the materials and methods section. This section can be vastly improvised (Example: It is okay to refer to previous articles for experiment steps but authors must mention the key details of methodology/conditions used for all experiments). The key details are missing for almost all experimental methods used in the study such as section # 2.4, 2.6, 2.7, 2.8, 2.10, 2.11, 2.12 so on and so forth for rest of sections and the lack of these details makes it difficult to assess the results as well. 

3) Did authors examine the effect of haloperidol treatment on EC progression (in vivo studies) using other human endometrial endothelial cells (HEEC) as mentioned in results Ln # 301 and 302. If not please avoid the use of HEEC and mention the cell line used to establish the model used in the study.

4) There is lot of inconsistency in the choice of cell lines or experiments performed. Example for data shown from different experiments in Figure 3, the authors have used three different cell lines or two cell lines and did not explain rational for choice of cell lines for respective experiments. If authors have chosen to show data from particular cell lines only, the authors must explain the rational and consider to include the data in supplemental data.

5) The authors are requested to carefully review all the data and label the figures/data with names of genes or proteins being measured/shown in the figures. The labels on figures must allow the reader to understand the data rather than referring to legends or text for every panel of the figure. Could authors please fix the issue. 

6) It is understandable from the data but the authors should consider to include the rational for selection of HEC1A cell line for in-vivo studies in results section.

7) Could authors please explain the rational for using different concentrations for in-vitro experiments. Example: For all cellular assays the authors have used 10 μM of haloperidol but for soft agar assay the authors have used 100μM of haloperidol? Did authors use lower concentration (10μM) of haloperidol for soft agar colony formation assay? If so please include the data.

8) The authors must include details of kits/panels used for the experiments in the study. Example: The authors have stated they have used RT2 Profile PCR 387 ArrayTM. If the catalog number is not provided it is difficult to understand what genes are being analyzed and to make sense of reported results. 

Minor concerns:

1) The authors have mentioned "150 μg of cell extracts" was used to probe for CSF-1 and phospho-Ser536-p65 by western blotting. Did authors use 150ug of total protein for western blot? 

2) Could authors please use standard acronyms (Example: Dopamine D2 receptor typically abbreviated as dopamine D2R, DRD2, D2DR, or D2R but not DD2R).

3) Could authors please clarify what do they mean by DR2R? If authors meant to say DRD2, please correct it accordingly. 

Author Response

Dear Reviewer,

            We are grateful to receive your exhaustive review. As listed below, we have assessed all of your comments and revised corresponding sections or written our explanations to each of your concerns. We hope these responses address your previous questions, but please do not hesitate to inquire us if any doubt lingers further.

Sincerely yours,

Chia-Hung Hsieh, Ph.D.
Professor
Graduate Institute of Biomedical Sciences
China Medical University
Taichung, Taiwan.
Tel: 886-4-22052121#7712 (office ) or 7717 (lab)
Fax: 886-4-22333641
E-mail: chhsiehcmu@mail.cmu.edu.tw
Office Address: No.91 Hsueh-Shih Road, Taichung, Taiwan 40402.

Reviewer’s Major Concerns

  1. Could authors please consider to elaborate the introduction especially on studies implicating antipsychotics as a risk factor for endometrial cancer. The authors have merely mentioned it and brief elaboration on this topic is required for the introduction.

Response: We sincerely appreciate your suggestion. The last paragraph of the introduction has been revised to include more details on the risk of antipsychotics in endometrial cancer. 

  1. Could authors please rewrite the materials and methods section. This section can be vastly improvised (Example: It is okay to refer to previous articles for experiment steps but authors must mention the key details of methodology/conditions used for all experiments). The key details are missing for almost all experimental methods used in the study such as section # 2.4, 2.6, 2.7, 2.8, 2.10, 2.11, 2.12 so on and so forth for rest of sections and the lack of these details makes it difficult to assess the results as well.

Response: We sincerely appreciate your comprehensive advice. We have fully revised sections # 2.4, 2.5, 2.6, 2.7, 2.8, 2.10, 2.11, 2.12, 2.13, 2.14, 2.16, and 2.17 to cover the major concepts and steps of all of our methods.

  1. Did authors examine the effect of haloperidol treatment on EC progression (in vivo studies) using other human endometrial endothelial cells (HEEC) as mentioned in results Ln # 301 and 302. If not please avoid the use of HEEC and mention the cell line used to establish the model used in the study.

Response: We sincerely appreciate your scrutiny. This appears to be a writing error that we have corrected from HEEC to HEC1A in the latest manuscript.

  1. There is lot of inconsistency in the choice of cell lines or experiments performed. Example for data shown from different experiments in Figure 3, the authors have used three different cell lines or two cell lines and did not explain rational for choice of cell lines for respective experiments. If authors have chosen to show data from particular cell lines only, the authors must explain the rational and consider to include the data in supplemental data.

Response: We sincerely acknowledge your concern. For the cell line choices of HECCs, HEC1A was categorized into type 1 endometrial carcinoma cell lines (well-differentiated and primary) while AN3CA and KLE into type 2 (poorly-differentiated and metastatic) according to published articles [1-3]. Therefore, some in vitro studies that only included HEC1A and AN3CA (Fig. 3G, 3H, 4C, 4D, 4J, 4K, 5C – 5G) represent two different types. Each experiment contains at least two cell lines to rule out the concern about cell-line-dependent biases.

  1. The authors are requested to carefully review all the data and label the figures/data with names of genes or proteins being measured/shown in the figures. The labels on figures must allow the reader to understand the data rather than referring to legends or text for every panel of the figure. Could authors please fix the issue.

Response: We sincerely appreciate your advice. We have reviewed all the figure labels as well as legends, and updated labels in Fig. 4 to present the names of the targets.

  1. It is understandable from the data but the authors should consider to include the rational for selection of HEC1A cell line for in-vivo studies in results section.

        Response: We sincerely acknowledge your concern. We opt for HEC1A due to several reasons. First, type 1 endometrial carcinoma responds well to the hyper-estrogenic condition [1], meaning that it may be more prone to the fluctuation of hormone or cytokine levels than type 2 endometrial carcinoma cell lines. This could be crucial since our study involved the influence of CSF-1. Second, early research demonstrated that antipsychotics may increase the risk of cancer via elevating prolactin level [4-6], which suggests that type 1 endometrial carcinoma could be more sensitive to haloperidol’s effects than type 2 endometrial carcinoma. Last but not least, our in vitro data suggested that haloperidol induces endometrial proliferation, carcinogenesis, and migration/invasion, which revealed that haloperidol may be significant to endometrial cancer development across both primary and metastatic stages. Therefore, we choose HEC1A as our in vivo model following a published article [7] to fully observe how haloperidol can influence endometrial cancer development from primary on-site proliferation to metastatic off-site growth, with consideration of the purported mechanism in haloperidol-induced endometrial carcinogenesis.

  1. Could authors please explain the rational for using different concentrations for in-vitro experiments. Example: For all cellular assays the authors have used 10 μM of haloperidol but for soft agar assay the authors have used 100μM of haloperidol? Did authors use lower concentration (10μM) of haloperidol for soft agar colony formation assay? If so please include the data.

Response: We sincerely appreciate your detailed inspection. This appeared to be a writing error which we have revised in section# 2.13. As shown in Fig. 3B, we actually tested four different concentrations of haloperidol – 0, 10, 50, and 100 μM –in the soft agar colony formation assay. Hence, both the cellular assays and soft agar colony formation assay tested 10 μM of haloperidol.

  1. The authors must include details of kits/panels used for the experiments in the study. Example: The authors have stated they have used RT2 Profile PCR 387 ArrayTM. If the catalog number is not provided it is difficult to understand what genes are being analyzed and to make sense of reported results.

Response: We sincerely appreciate your experienced recommendation. We have included the ten signal transduction pathways that are analyzed in the kit in addition to its catalog number.

Reviewer’s Minor Concerns

  1. The authors have mentioned "150 μg of cell extracts" was used to probe for CSF-1 and phospho-Ser536-p65 by western blotting. Did authors use 150ug of total protein for western blot?

Response: We sincerely appreciate your attention to detail. This appeared to be another writing error that we have carefully reviewed our laboratory protocols and revised in section# 2.5. We added 30 μg of protein samples into each well of stacking gel.  

  1. Could authors please use standard acronyms (Example: Dopamine D2 receptor typically abbreviated as dopamine D2R, DRD2, D2DR, or D2R but not DD2R)

Response: We sincerely appreciated your scientific caliber. All the acronyms for Dopamine D2 receptor have been revised to DRD2 according to NCBI Gene. Moreover, all the spelling of genes and proteins have been reviewed and properly italicized for the gene names.

  1. Could authors please clarify what do they mean by DR2R? If authors meant to say DRD2, please correct it accordingly.

Response: We sincerely appreciated your scientific caliber. We have corrected DR2R into DRD2.

Citations

  1. Theisen, E.R.; Gajiwala, S.; Bearss, J.; Sorna, V.; Sharma, S.; Janat-Amsbury, M. Reversible inhibition of lysine specific demethylase 1 is a novel anti-tumor strategy for poorly differentiated endometrial carcinoma. BMC Cancer 2014, 14, 752, doi:10.1186/1471-2407-14-752.
  2. Noumoff, J.; Haydock, S.W.; Sachdeva, R.; Heyner, S.; Pritchard, M.L. Characteristics of cell lines derived from normal and malignant endometrial tissue. Gynecol Oncol 1987, 27, 141-149, doi:10.1016/0090-8258(87)90286-1.
  3. Wang, Y.; Yang, D.; Cogdell, D.; Hu, L.; Xue, F.; Broaddus, R.; Zhang, W. Genomic characterization of gene copy-number aberrations in endometrial carcinoma cell lines derived from endometrioid-type endometrial adenocarcinoma. Technol Cancer Res Treat 2010, 9, 179-189, doi:10.1177/153303461000900207.
  4. Yamazawa, K.; Matsui, H.; Seki, K.; Sekiya, S. A case-control study of endometrial cancer after antipsychotics exposure in premenopausal women. Oncology 2003, 64, 116-123, doi:10.1159/000067769.
  5. Ramírez-de-Arellano, A.; Villegas-Pineda, J.C.; Hernández-Silva, C.D.; Pereira-Suárez, A.L. The Relevant Participation of Prolactin in the Genesis and Progression of Gynecological Cancers. Frontiers in Endocrinology 2021, 12, 747810, doi:10.3389/fendo.2021.747810.
  6. Petty, R.G. Prolactin and antipsychotic medications: mechanism of action. Schizophrenia Research 1999, 35 Suppl, S67-73, doi:10.1016/s0920-9964(98)00158-3.
  7. Haldorsen, I.S.; Popa, M.; Fonnes, T.; Brekke, N.; Kopperud, R.; Visser, N.C.; Rygh, C.B.; Pavlin, T.; Salvesen, H.B.; McCormack, E.; et al. Multimodal Imaging of Orthotopic Mouse Model of Endometrial Carcinoma. PloS One 2015, 10, e0135220, doi:10.1371/journal.pone.0135220.

Reviewer 2 Report

The manuscript presents a detailed investigation of the effect of haloperidol on endometrial cancer (EC) cell progression and invasion and survival in mice with orthotopic EC. The clarity in the writing of the manuscript and in the presentation of methods and results is noteworthy. In particular, the introduction is very clear and perfectly focused on the role that haloperidol has been shown to have in various types of cancer, with a correct and very well summarized review of the literature. As for the section on materials and methods, they describe in detail each of the methods used during the development of their research. In addition, the numerous methods used and the fact that data from both in vitro and in vivo assays are provided, as well as clinical results, stand out. In results and discussion, very clear and very well justified each of the results that corroborate at all times the conclusion reached by the authors that haloperidol is a carcinogenic compound for EC.

Author Response

Dear Reviewer,

            We are sincerely grateful for your kind comments. Please don’t hesitate to inform us if any questions surface.

Sincerely yours,

Chia-Hung Hsieh, Ph.D.
Professor
Graduate Institute of Biomedical Sciences
China Medical University
Taichung, Taiwan.
Tel: 886-4-22052121#7712 (office ) or 7717 (lab)
Fax: 886-4-22333641
E-mail: chhsiehcmu@mail.cmu.edu.tw
Office Address: No.91 Hsueh-Shih Road, Taichung, Taiwan 40402.

Round 2

Reviewer 1 Report

The authors have addressed most of the concerns and are requested to address following concerns before the manuscript could be considered for publication.

Major concerns:

1) Could authors please clarify what does authors mean by "The resulting cDNAs were diluted to the same concentration measured by a spectrophotometer" in Section # 2.6? 

2) The authors are supposed to include the concentration of primers used rather than mentioning the volume of primers. Please fix the issue on Ln # 209 of page # 4.

3) The authors were asked to include key details on the conditions used for respective experiments mentioned in earlier comment # 2 but currently each section of methods were edited to include very basic details of the experiments as well. The authors can either keep these basic details or can re-edit these sections and keep the key details such as number of cells, concentrations of protein/RNA/primers, volume of samples, and route of drugs administration for respective experiments.  

Minor concerns:

1) Could authors please correct "dopamine receptors" to "dopamine receptor" on Ln # 67 of page # 2.

2) Please replace "secary" with "secondary" on Ln # 191 of page # 4.

3) Please replace "secary" with "secondary" on Ln # 228 of page # 4.

Author Response

Dear Reviewer 1,

We are sincerely grateful for your reviews. A point-by-point response letter has been listed below. We hope these amendments and explanations can clear out your concerns. Again, please let us know if any more questions surface.

Sincerely yours,

Chia-Hung Hsieh, Ph.D.
Professor
Graduate Institute of Biomedical Sciences
China Medical University
Taichung, Taiwan.
Tel: 886-4-22052121#7712 (office ) or 7717 (lab)
Fax: 886-4-22333641
E-mail: chhsiehcmu@mail.cmu.edu.tw
Office Address: No.91 Hsueh-Shih Road, Taichung, Taiwan 40402.

Major concerns

  1. Could authors please clarify what does authors mean by "The resulting cDNAs were diluted to the same concentration measured by a spectrophotometer" in Section # 2.6?

Response: We thank you for the inquiry. We measured RNA concentrations before cDNA transcription and cDNA concentrations before Q-PCR. As this description created considerable confusion, we have removed it in our latest manuscript and included the cDNA concentration (50 ng per 20 μL) for Q-PCR reaction.

  1. The authors are supposed to include the concentration of primers used rather than mentioning the volume of primers. Please fix the issue on Ln # 209 of page # 4.

Response: We thank you for the attention to detail. We have included the primer concentration (10 μM) for the Q-PCR’s method in our latest manuscript.

  1. The authors were asked to include key details on the conditions used for respective experiments mentioned in earlier comment # 2 but currently each section of methods were edited to include very basic details of the experiments as well. The authors can either keep these basic details or can re-edit these sections and keep the key details such as number of cells, concentrations of protein/RNA/primers, volume of samples, and route of drugs administration for respective experiments.

Response: We thank you for your suggestion. We have added the concentration of cDNA and primers in the Q-PCR section as well as cDNA’s concentration in the Gene expression profiling assay section. We also reviewed again to ensure that the essential parts, such as the number of cells, concentrations of protein/RNA/cDNA/primers, the volume of samples, the route of drug administration, and particularly those differing from the manufacturers’ instructions, have been appropriately incorporated in our method section.

Minor concerns

  1. Could authors please correct "dopamine receptors" to "dopamine receptor" on Ln # 67 of page # 2.
  2. Please replace "secary" with "secondary" on Ln # 191 of page # 4.
  3. Please replace "secary" with "secondary" on Ln # 228 of page # 4.

Response to all the minor concerns: We thank you again for the attention to detail. All of the typos have been corrected in our latest manuscript.